# Cryo-EM structure of alpha-synuclein fibrils

**Ricardo Guerrero-Ferreira[1], Nicholas MI Taylor[1†], Daniel Mona[2], Philippe Ringler[1], Matthias E Lauer[3], Roland Riek[4], Markus Britschgi[2], Henning Stahlberg[1]\***

[1]Center for Cellular Imaging and NanoAnalytics, Biozentrum, University of Basel, Basel, Switzerland; [2]Roche Pharma Research and Early Development, Neuroscience, Ophthalmology and Rare Diseases Discovery and Translational Area/Neuroscience Discovery, Roche Innovation Center Basel, Basel, Switzerland; [3]Roche Pharma Research and Early Development, Chemical Biology, Roche Innovation Center Basel, Basel, Switzerland; [4]Laboratory of Physical Chemistry, ETH Zürich, Zürich, Switzerland

**Abstract** Parkinson's disease is a progressive neuropathological disorder that belongs to the class of synucleinopathies, in which the protein alpha-synuclein is found at abnormally high concentrations in affected neurons. Its hallmark are intracellular inclusions called Lewy bodies and Lewy neurites. We here report the structure of cytotoxic alpha-synuclein fibrils (residues 1–121), determined by cryo-electron microscopy at a resolution of 3.4 Å. Two protofilaments form a polar fibril composed of staggered β-strands. The backbone of residues 38 to 95, including the fibril core and the non-amyloid component region, are well resolved in the EM map. Residues 50–57, containing three of the mutation sites associated with familial synucleinopathies, form the interface between the two protofilaments and contribute to fibril stability. A hydrophobic cleft at one end of the fibril may have implications for fibril elongation, and invites for the design of molecules for diagnosis and treatment of synucleinopathies.
DOI: https://doi.org/10.7554/eLife.36402.001

**\*For correspondence:**
Henning.Stahlberg@unibas.ch

**Present address:** †Structural Biology of Molecular Machines Group, Protein Structure and Function Programme, Novo Nordisk Foundation Center for Protein Research, Faculty of Health and Medical Sciences, University of Copenhagen, Copenhagen, Denmark

## Introduction

Parkinson's disease (PD) is a neurodegenerative disorder characterized by the presence of Lewy bodies (LB) and Lewy neurites (LN). *Spillantini et al. (1997)* identified fibrils formed by the presynaptic protein alpha-synuclein (α-Syn, 140 residues, ~14 kD) as the main component of these human brain inclusions (*Spillantini et al., 1998*; *Spillantini et al., 1997*).

Certain α-Syn fibril forms can seed LB-like and LN-like inclusions in cell culture and intra-neuronal aggregation of mouse α-Syn in vivo (*Luk et al., 2009*; *Thakur et al., 2017*; *Volpicelli-Daley et al., 2014*). In addition, abnormal α-Syn produces neuronal cell inclusions and axonal spheroids, as well as oligodendrocytic aggregates, known as glial cytoplasmic inclusions, found abundantly in Multiple System Atrophy (MSA) (*Arima et al., 1998*; *Tu et al., 1998*), which makes α-Syn fibrils an important target for the development of diagnostic tools and therapeutic strategies for PD and related synucleinopathies.

Despite α-Syn fibrils, other forms of α-Syn might also be involved in neurodegeneration, such as an oligomeric α-Syn intermediate (*Danzer et al., 2007*; *Lashuel et al., 2002*; *Outeiro et al., 2008*; *Vicente Miranda et al., 2017*; *Villar-Piqué et al., 2016*; *Winner et al., 2011*), or the process of fibril aggregation itself (*Oueslati et al., 2010*; *Reynolds et al., 2017*; *Taschenberger et al., 2012*). Fibrils of α-Syn show significant fibril strain polymorphism (*Peelaerts et al., 2015*).

Several factors point to α-Syn as an important player in the onset of PD: (i) six known point mutations in the α-Syn gene (SNCA) are associated with familial forms of synucleinopathies: A30P

**eLife digest** People with Parkinson's disease have damaged cells in a part of the brain involved in movement, learning and reward-seeking behaviors. These cells contain blob-like aggregates that contain abnormally high amounts of a protein called alpha-synuclein. It is generally believed that, within these blobs, this protein clusters together into small needles called fibrils.

Discerning the structure of a fibril could help researchers to understand both how alpha-synuclein damages brain cells and how diseases like Parkinson's spread. Biophysicists have attempted to reveal the fibril structure previously. But many of these efforts only looked at short segments of the alpha-synuclein protein. Researchers still need more detailed imagery of the fibrils to confirm previous findings regarding their architecture and ultimately to identify ways to counteract the damage they cause.

Guerrero-Ferreira et al. used a technique called cryo-electron microscopy to capture images of frozen fibrils made from a version of human alpha-synuclein that readily aggregates and that is only slightly shorter than the full-length protein. Processing these high-resolution images with computer software then revealed a three-dimensional model of the fibril structure, in which fine details are clearly visible. In the fibril, the proteins cluster to form a helix, similar to a flight of stairs. Each turn of the helix is formed by two alpha-synuclein molecules, facing each other but rotated by almost 180 degrees from one another. The three-dimensional model displays which parts of the protein lie at the core of the helix and thereby stabilize the fibril structure. Guerrero-Ferreira et al. speculate that fibrils may also take alternative forms because common alpha-synuclein mutations, which correlate with disease, would destabilize the observed helical structure.

In the future, researchers may be able to use the features of this three-dimensional model to help design molecules that would make the fibrils detectable via medical imaging. This could help doctors to diagnose people with Parkinson's disease at an earlier stage. Further research is also needed to understand where and how fibrils form, if differences in fibril structures exist within or between patients, possibly leading to different sub-classes of the disease, and how such fibrils interact with and possibly damage human brain cells.

DOI: https://doi.org/10.7554/eLife.36402.002

(*Krüger et al., 1998*), E46K (*Zarranz et al., 2004*), H50Q (*Appel-Cresswell et al., 2013*), G51D (*Lesage et al., 2013*), A53E (*Pasanen et al., 2014*), and A53T (*Polymeropoulos et al., 1997*); (ii) animal models suggest a role of α-Syn in the etiology of PD, Dementia with Lewy Bodies (DLB), and MSA (*Feany and Bender, 2000*; *Hashimoto et al., 2003*; *Periquet et al., 2007*; *Tyson et al., 2017*); (iii) individuals with duplications or triplications of the α-Syn gene exhibit overexpression of α-Syn and develop PD (*Ibáñez et al., 2004*; *Singleton et al., 2003*).

Two related proteins, β-synuclein (β-Syn) and γ-synuclein (γ-Syn), with sequence homology to α-Syn, have been described (*Clayton and George, 1998*; *Jakes et al., 1994*; *Stefanis, 2012*). β-Syn and α-Syn share the greatest aminoacid sequence homology, with β-Syn lacking 12 amino acids (residues 71 to 82) within the non-amyloid component region (NAC; residues 61–95 in α-Syn) (*Giasson et al., 2001*; *Uéda et al., 1993*). In synucleins, regions with the highest homologies are located in the structurally heterogeneous, amino-terminal half (residues 10–84 in α-Syn) composed of 5 to 6 imperfect repeats with the consensus sequence KTKEGV (*Der-Sarkissian et al., 2003*). In contrast, the carboxyl terminus is highly negatively charged and unstructured (*Chen et al., 2007*; *Vilar et al., 2008*).

A number of post-translational modifications have been described for α-Syn including phosphorylation (*Anderson et al., 2006*; *Fujiwara et al., 2002*; *Paleologou et al., 2010*), acetylation (*Iyer et al., 2016*; *Maltsev et al., 2012*), ubiquitination (*Hasegawa et al., 2002*), and C-terminal truncation (*Anderson et al., 2006*; *Crowther et al., 1998*). C-terminal truncation of α-Syn occurs normally in vivo, under physiological conditions and it has been shown to promote fibrillization (*Crowther et al., 1998*; *Li et al., 2005*; *Liu et al., 2005*; *Wang et al., 2016*). In turn, truncated forms of α-Syn play a role in inducing Lewy body formation (*Dufty et al., 2007*; *Li et al., 2005*; *Prasad et al., 2012*), suggesting that truncation by proteolysis may be important in the pathological process.

In vivo studies investigating α-Syn aggregation demonstrated that activation of the inflammasome and more specifically caspase-1, the enzymatic component of the inflammasome, leads to the production of an α-Syn fragment truncated at aspartic acid 121 (D121) (*Wang et al., 2016*). This C-terminally-truncated α-Syn form (α-Syn(1-121)) aggregates more rapidly than full-length α-Syn (including disease-associated mutants), and its production is associated with cell toxicity. Furthermore, the use of VX-765, a pro-drug that produces a specific inhibitor of caspase-1 in vivo (*Wannamaker et al., 2007*), improved survival of a neuronal cell model of PD (*Wang et al., 2016*), and reduced neurodegeneration in a transgenic mouse model of MSA (*Bassil et al., 2016*), suggesting an important role of α-Syn(1-121) in cellular toxicity in both, cell cultures as well as a mouse model.

To this date, high resolution structures of α-Syn fibrils are limited to the results of a micro-electron diffraction (microED) study of two small segments of the protein (*Rodriguez et al., 2015*) and a solid-state NMR structure obtained from ~5 nm diameter, single protofilaments (*Tuttle et al., 2016*), in addition to solid state NMR studies at the secondary structure level (*Bousset et al., 2013*; *Kim et al., 2009*; *Vilar et al., 2008*), and X-ray diffraction studies of shorter segments of α-Syn (*Li et al., 2014*), or α-Syn bound to other molecules (*De Genst et al., 2010*; *Gruschus et al., 2013*; *Rao et al., 2010*; *Ulmer et al., 2005*; *Xie et al., 2010*; *Yagi-Utsumi et al., 2015*; *Zhao et al., 2011*).

Here, we report the atomic structure of α-Syn(1-121) fibrils determined by cryo-electron microscopy (cryo-EM). The structure allows conclusions about the organization of α-Syn fibrils at near-atomic resolution, suggest mechanisms for fibril formation and growth, and allows conclusions on fibril stability.

## Results and discussion

### The 3D structure of α-Syn amyloid fibrils

Several preparations of recombinant human α-Syn fibril were screened by negative stain transmission electron microscopy (TEM; *Figure 1—figure supplement 1*). These included fibrils formed by full length α-Syn (*Figure 1A*), α-Syn phosphorylated at serine 129, N-terminally acetylated, and C-terminal truncated α-Syn comprised of residues 1–119 (α-Syn(1-119)), 1–121 (α-Syn(1-121)), or 1–122 (α-Syn(1-122)).

The diameters of the α-Syn fibrils produced varied from 5 nm to approximately 10 nm when studied by negative stain TEM. The fibrils formed by α-Syn(1-121) were straight, between 20 and 500 nm long and the only ones of consistent diameters of 10 nm (*Figure 1B*, *Figure 1—figure supplement 1E*). This fibrillar form α-Syn(1-121) has been described as an aggregation-prone species resulting from α-Syn truncation by caspase-1 (*Wang et al., 2016*). The recombinantly produced α-Syn(1-121) used here showed a similarly aggressive aggregation profile.

Preparations of α-Syn(1-121) fibrils were quick-frozen in the holes of fenestrated carbon coated cryo-electron microscopy (cryo-EM) grids, and imaged with a Titan Krios 300kV cryo-EM instrument, equipped with a Quantum-LS energy filter and a K2 Summit direct electron detector. Helical image processing of recorded cryo-EM movies produced a 3D reconstruction of the α-Syn(1-121) fibril at an overall resolution of 3.4 Å (*Figure 1C and D*, *Figure 1—figure supplement 2*, *Figure 2*, and *Video 1*).

Our 3D map shows that fibrils are formed by two protofilaments, each of 5 nm in diameter (*Figure 1*). These lack C2 symmetry, but are related by an approximate $2_1$ screw symmetry, akin to the symmetry exhibited by the paired helical filaments of tau (*Fitzpatrick et al., 2017*) and by amyloid-ß(1-42) filaments (*Gremer et al., 2017*). α-Syn(1-121) fibrils are therefore polar, meaning that both protofibrils are aligned into the same direction. The position of a given ß-sheet in a protofilament is produced by the rotation of 179.5° of one sheet around its axis (helical twist), followed by a vertical translation of 2.45 Å (helical rise). This ß-sheet arrangement results in a spacing of 4.9 Å between α-Syn subunits in successive rungs of a single protofilament (*Figure 1C and D*). The quality of the EM map allowed an atomic model of the region between residues L38 and V95 to be built.

Each α-Syn(1-121) molecule comprises eight in-register parallel β-strands (i.e. residues 42–46 (β1), 48–49 (β2), 52–57 (β3), 59–66 (β4), 69–72 (β5), 77–82 (β6), 89–92 (β7), and 94-(~102) (β8)), which are interrupted by glycine residues (i.e. G41 before β1, G47 between β1 and β2, G51 between β2 and β3, G67 and G68 between β4 and β5, G73 between β5 and β6, G84 and G86 between β6 and β7,

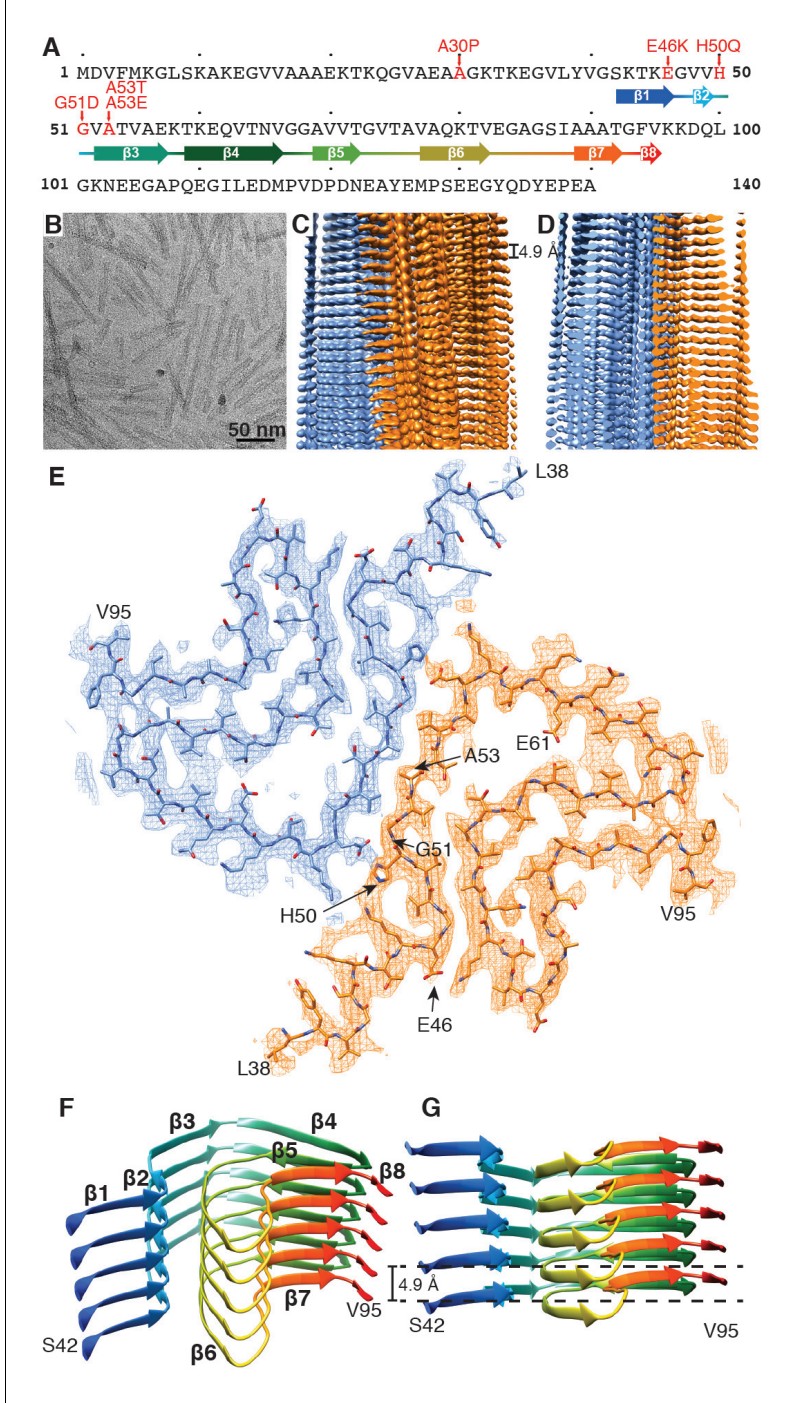

**Figure 1.** Structure of α-Syn(1-121) fibril. (**A**) Schematic depicting the sequence of human α-Syn. The positions of the known familial mutations are indicated. β-strand regions are indicated by arrows colored from blue to red. (**B**) Cryo-EM micrograph depicting the distribution and general appearance of α-Syn fibrils. (**C**) Cryo-EM reconstruction of α-Syn(1-121) fibrils showing two protofilaments (orange and blue). (**D**) Cross-section of (**C**) illustrating the clear separation of the β-strands, also shown in *Figure 1—figure supplement 3A and B*. (**E**) Cross-section of a fibril (along the axis) illustrating the arrangement of the two protofilaments (orange and blue) and fitted atomic model. Positions of the initial (L38) and final (V95) residues fitted are indicated, as well as the initial and final residue of the NAC region (E61 to V95). Arrows indicate the location of four of the five α-Syn residues where familial mutations associated with PD occur. (**F**) Distribution of β-strands in a single protofilament of the α-Syn fibril, corresponding to residues 42 to 95. Color scheme, as in (**A**). (**G**) As in (**F**) but a perpendicular view to the fibril axis illustrating height differences in some areas of a single protofilament.

*Figure 1 continued on next page*

*Figure 1 continued*

DOI: https://doi.org/10.7554/eLife.36402.003

The following figure supplements are available for figure 1:

**Figure supplement 1.** Negative stain TEM images of α-Syn strains.

DOI: https://doi.org/10.7554/eLife.36402.004

**Figure supplement 2.** Local resolution estimation and FSC curves.

DOI: https://doi.org/10.7554/eLife.36402.005

**Figure supplement 3.** Details of atomic model and density.

DOI: https://doi.org/10.7554/eLife.36402.006

and G93 between β7 and β8) or an arch (i.e. E57-K58 between β3 and β4) (*Figure 1A,F and G*). The β-strands β2-β7 wind around a hydrophobic intra-molecular core composed of only alanine and valine residues and one isoleucine (i.e. V48, V49, V52, A53, V55, V63, A69, V70, V71, V74, A76, V77, A78, I88, A89, A90, A91). Considering that these hydrophobic clusters are maintained along the fibril, they are likely to contribute to the stability of the protofilament. The hydrophobic core is surrounded by two hydrophilic regions (i.e. (i): Q79, T81, and (ii): T72, T75, T54, T59, and E61) both still within the core of the structure (*Figure 3*). While most of these side chains form so-called side chain

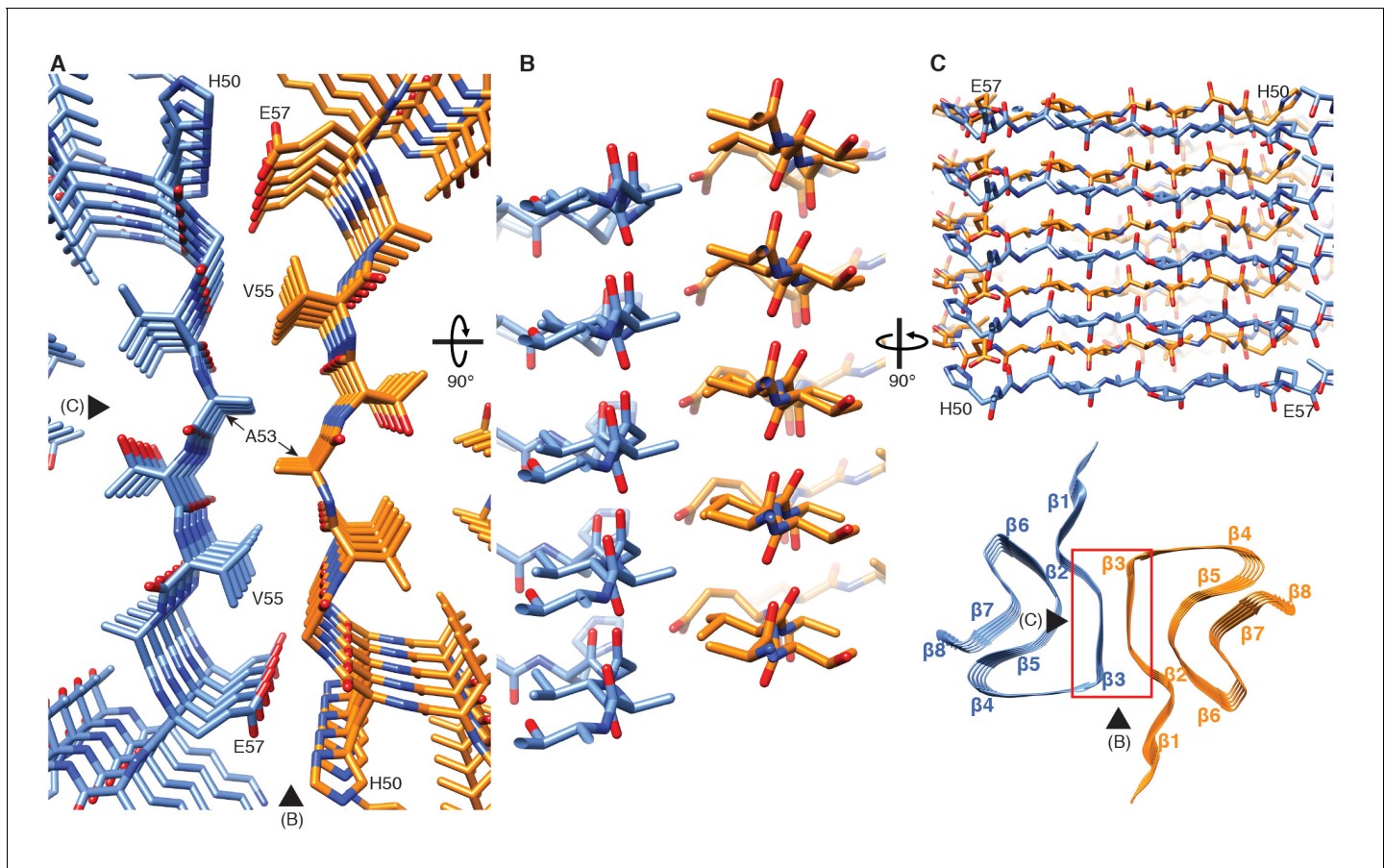

**Figure 2.** Interface region between two protofilaments of the α-Syn(1-121) fibril. (**A**) View along the axis of the fibril as indicated by the red rectangle on the ribbon diagram (bottom right). (**B**) (**C**) Side views of the fibril with orientations indicated by arrowheads in (**A**) and the ribbon diagram (bottom right). Panels (**B**) and (**C**) clearly illustrate the $2_1$ screw symmetry that results from the staggered arrangement of subunits.

DOI: https://doi.org/10.7554/eLife.36402.007

The following figure supplement is available for figure 2:

**Figure supplement 1.** Stacking of β-strands.

DOI: https://doi.org/10.7554/eLife.36402.008

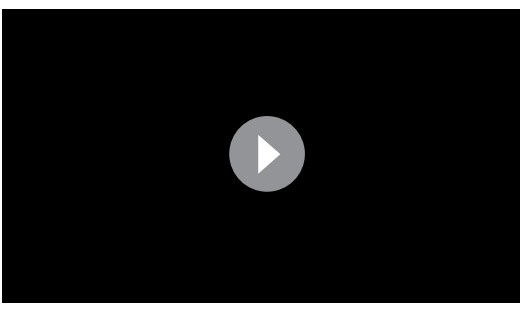

**Video 1.** Cryo-EM structure of alpha-synuclein fibril. Details of the cryo-EM reconstruction of an alpha-synuclein fibril at 3.4 Å resolution, illustrating the interaction between two protofilaments, the 4.9 Å spacing between β-strands of a single protofilament and monomer topology in the protofilament core. DOI: https://doi.org/10.7554/eLife.36402.009

hydrogen bond ladders (*Nelson et al., 2005*; *Riek, 2017*), the second hydrophilic region comprising four threonine residues and a negatively charged glutamic acid side chain surrounds a tunnel filled with some ordered molecules of unknown nature, as evidenced by an additional density (*Figure 1—figure supplement 3D*). The less well defined β1 and β8 strands are attached to the core, while the first 37 N-terminal residues and the last ~20 C-terminal residues of α-Syn(1-121) are not visible in the 3D reconstruction (*Figure 1E* and *Figure 1—figure supplement 2A*), indicating a disordered structure in line with quenched hydrogen/deuterium exchange – solution-state NMR (H/D exchange NMR) and limited proteolysis (*Vilar et al., 2008*), which showed these terminal segments to be unprotected in nature. Together with our results, this suggests that approximately 40 residues of both the N- and C-terminal ends of full-length human α-Syn

are flexible, and surround the structured core of the fibril with a dense mesh of disordered tails, similar to the 'fuzzy coat' recently described in the cryo-EM tau structure (*Fitzpatrick et al., 2017*).

Two β-sheets (one from each protofilament) interact at the fibril core via a hydrophobic steric zipper-geometry comprised of β-strand β3 (i.e. residues G51-A56). As a consequence, two α-Syn molecules per fibril layer are stacked along the fibril axis (*Figure 2B and C*). The side chains of residues A53 and V55 form the inter-molecular surface contributing to the interface between the two protofilaments, which is further stabilized by a surface-exposed salt bridge between E57 and H50 that might be sensitive to pH, as an unprotected histidine has a pK of ~6.2 (*Figure 1—figure supplement 3H*). The same structure with a steric zipper topology was found in micro-crystals of the peptide comprising residues G47-A56 (*Rodriguez et al., 2015*). Interestingly, the β-strand β6 that is sandwiched between β-strands β2/β3 and β7 is also aligned with a neighboring molecule but shifted

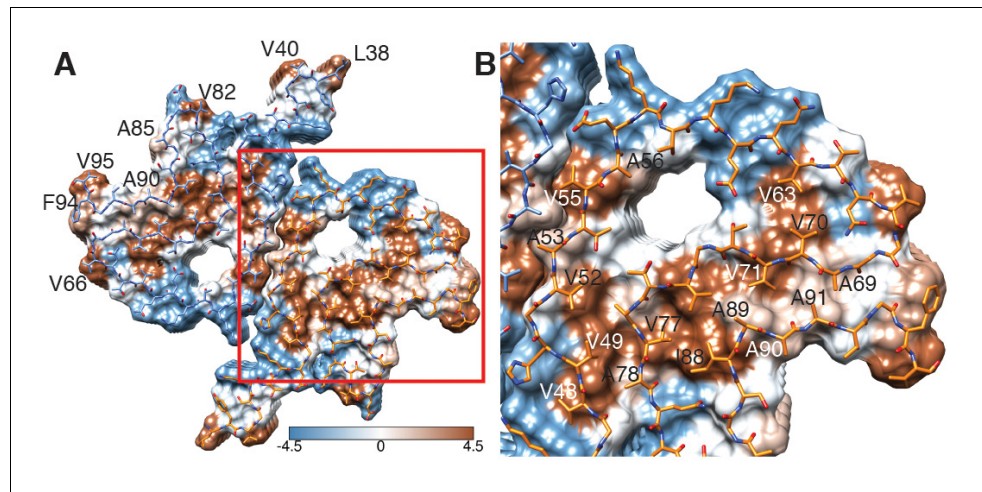

**Figure 3.** Hydrophobicity of α-Syn(1-121) fibrils. (**A**) Top view (fibril axis) of the hydrophobic regions (brown) in a fibril where the hydrophobic pocket at the interface between two protofilaments is evident. Hydrophobicity score from hydrophilic (−4.5, blue) to hydrophobic (4.5, brown) is indicated by the color bar. Hydrophobic residues on the outer surface of the fibril are indicated. (**B**) Close-up of the region highlighted in (**A**) indicating the hydrophobic core composed of alanines, valines and a single isoleucine (I88). Residues forming the hydrophilic region (blue) that surrounds the hydrophobic region of the core are also visible. DOI: https://doi.org/10.7554/eLife.36402.010

by one monomer along the fibril axis, as shown in *Figure 1G* and *Figure 2—figure supplement 1*. Thus, hetero and homo steric zippers are both present in the 3D structure. Of these, the homo steric zipper at the inter-molecular interface has an extensive and well-packed β-strand interface, forming a very densely packed fibril. This stacking generates an asymmetric fibril with two distinct ends. Furthermore, the hydrophobic core of the fibril is composed of β-strands that interact with each other in a half-stacked zipper topology, contrasting with the hydrophilic core comprised of β-strands β4 and β5, which are non-stacked (*Figure 1G* and *Figure 2—figure supplement 1*). The latter confirms previous results from site-directed spin labeling experiments, which show that the region including residues 62–67 at the beginning of the NAC region, has a pronounced lack of stacking interactions (*Chen et al., 2007*).

The outer surface of the ordered region of the fibrils is mostly hydrophilic, with a few exceptions (i.e. L38, V40, V82, A85, A90, F94, V95) (*Figure 3A*). The side chain of V66 should probably not be classified as surface exposed because of its interaction with β-strand β8 (*Figure 1—figure supplement 2A*). If we ignore the influence of the non-polar alanine residues due to the small size of their side chains, the surface of the fibrils has two highly hydrophobic regions formed by residues L38 and V40, and by residues F94 and V95. Other interesting properties of the surface are the salt bridge formed by the side chains of E46 and K80 (*Figure 1—figure supplement 3G*) and the rather highly positive clustering of K43, K45, K58, H50 that requests the binding of a counter-ion, as it is supported by an observed density (*Figure 1—figure supplement 3C*).

## The familial PD mutations in the context of the 3D fibril structure

Six familial mutations in α-Syn are known to be associated with PD and other synucleinopathies (i. e. A30P, E46K, H50Q, G51D, A53E, and A53T). Of these, all but A30P are located in the heart of the core of the fibril structure presented here (*Figure 1A and E*). E46 forms a salt bridge with K80 (*Figure 1—figure supplement 3G*). The mutation of the glutamic acid E46 to a positively charged lysine in an E46K mutant would thus induce a charge repulsion between β-strands β1 and β6, likely destabilizing this α-Syn fibril structure (*Tuttle et al., 2016*). The familial PD/DLB-causing mutation E46K was found to enhance phosphorylation in mice (*Mbefo et al., 2015*), and its toxic effect was increased by the triple-K mutation (E35K, E46K, E61K) in neuronal cells (*Dettmer et al., 2017*).

Previous high-resolution structures of α-Syn only included small peptides or single protofilaments (*Rodriguez et al., 2015*; *Tuttle et al., 2016*). Our 3D map suggests structural contributions of some familial mutations to fibril stability, since H50, G51 and A53 are all involved in the inter-molecular contact between the two β-sheets from adjacent protofilaments at the core of the here studied α-Syn(1-121) fibrils. Mutation of the positively charged histidine 50 into a polar, uncharged glutamine in the H50Q mutant would likely interfere with the salt bridge established between residues E57 and H50 (*Figure 1—figure supplement 3H*). Adding to the absent side-chain of glycine 51 a negatively charged aspartic acid in mutant G51D, or transforming the small side-chain of alanine A53 into a larger threonine in mutant A53T, would likely disrupt the steric zipper interaction between the two protofibrils, whereby the A53T mutation would in addition change the highly hydrophobic surface at the zipper to partly hydrophilic one. In our α-Syn(1-121) fibril structure, A53 is part of a hydrophobic pocket that defines the interaction of protofilaments and likely contributes to fibril stability as the hydrophobic interactions exist along the fibril axis. Mutations at the core of this α-Syn fibril would compromise the formation of the structure presented here. This suggests that a different fibril structure (i.e. fibril strain) could be formed from α-Syn containing the above discussed familial PD mutations.

Several features of our structure, such as non-functional hydrophobic surface patches (*Figure 3*), a hydrophilic tunnel (*Figure 1—figure supplement 3D*), and a positively charged side chain arrangement like the one comprised of residues K43, K45, K58, H50 (*Figure 1—figure supplement 3C*) are not found in functional amyloid structures such as that of HET-s (*Wasmer et al., 2008*). However, similar structural characteristics have been previously observed for pathological tau filaments obtained from Alzheimer's disease brains where (i): lysine and tyrosine residues play a similarly stabilizing role in the interface region of two protofilaments of the straight filaments (SF), and (ii): the area in the center of the protofilaments is dominated by hydrophilic residues (*Fitzpatrick et al., 2017*). It is plausible that these structural features might arise because folding to form the amyloid fibril structure is dictated by the need to bury the maximum number of hydrophobic side-chains as efficiently as possible, as is also the case for the Aβ(1-42) amyloid fibrils (*Gremer et al., 2017*).

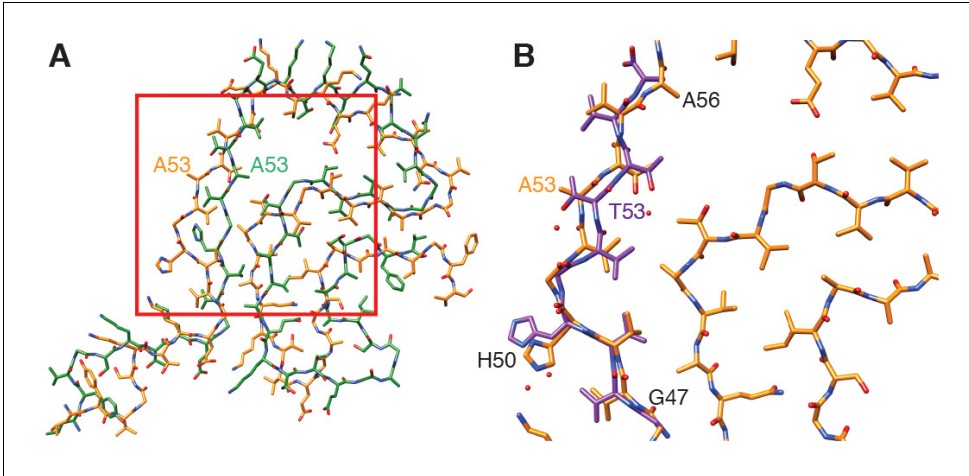

**Figure 4.** Comparison of α-Syn(1-121) fibrils with previous α-Syn fibril structures. (**A**) Overlay with the solid-state NMR structure from *Tuttle et al. (2016)* (green). Our α-Syn structure is orange in both overlays. (**B**) Overlay with the preNAC segment obtained by micro-ED by *Rodriguez et al. (2015)* (purple). The red square in (**A**) indicates the area of our structure shown in (**B**). Residue 53 is mutated (i.e. A53T) in the micro-ED structure.
DOI: https://doi.org/10.7554/eLife.36402.011

The artificial, highly toxic, but not synucleinopathy-related mutant E57K (*Winner et al., 2011*) is interesting to mention in the context of the 3D structure presented, because E57 is also at the inter-molecular interface (*Figure 2*). The presence of a positive lysine side chain at this position in the E57K mutant would significantly interfere with the formation of the interface and even the amyloid fibril (*Winner et al., 2011*). Indeed, this mutant was designed in a successful structure-based attempt to interfere with amyloid fibril formation (at least under some conditions) (*Winner et al., 2011*). Furthermore, both in a lentivirus-rat system as well as in a transgenic mouse model, the E57K mutant formed a significant amount of oligomers and was highly toxic, resulting in a large decay of TH-sensitive neurons in the *substantia nigra* of rats and a motor phenotype reminiscent of PD in mice (*Winner et al., 2011*). Thus, the artificial mutant E57K can be regarded as a 'familial PD-like' mutation both from the in vivo and from the structure/mechanism-based point of view.

## Comparison with earlier structural data

Full-length α-Syn subunits in a fibril studied by NMR ([*Tuttle et al., 2016*], PDB 2N0A) were found to be in a roughly similar secondary structure arrangement as in the here reported structure of α-Syn(1-121) (*Figure 4A*), even though the primary structure and the side-chain interactions of our here reported structure are very different from the NMR structure. Most importantly, the fibrils used for the NMR study were only approximately 5 nm wide, which corresponds to the diameter of a single protofilament. The larger diameter of our fibrils, 10 nm, results from the interaction between two protofilaments, which allowed us to hypothesize on the nature of α-Syn(1-121) protofilament interactions. Fibrils of 5 to 10 nm in diameter found in *substantia nigra* samples from the brain of PD patients, (*Crowther et al., 2000*), cingulate cortex of patients with DLB (*Spillantini et al., 1998*), cerebral cortex of PD patients (*Kosaka et al., 1976*), and in-vitro aggregated samples (*Bousset et al., 2013*). *Crowther et al. (2000)* had already suggested that the 10 nm filaments are the result of the interaction between 5 nm protofilaments.

An important difference between our here reported structure and the NMR structure reported by *Tuttle et al. (2016)* is the orientation of residue A53. The mutation A53T is associated with early onset PD. In our structure, residue A53 faces the interface between the two protofibrils and thereby likely contributes to fibril stability. In contrast, *Tuttle et al. (2016)* reported in their NMR structure A53 to point towards the hydrophobic core of the one observed individual protofilament, which may explain the lack of 10 nm fibrils in their sample. However, it is also noted here that the NMR study by *Tuttle et al. (2016)* showed a significant disagreement among the ten lowest-energy NMR structures for residues 51–67 [Figure 3d in *Tuttle et al. (2016)*], indicating a lower confidence for those

residues in the NMR structure. Our here reported cryo-EM map has the side-chains for those residues pointing into the opposite direction as reported in the *Tuttle et al. (2016)* structure.

Our structure includes a serine residue at position 87 (*Figure 1—figure supplement 3E*), which is one of the several phosphorylation sites in α-Syn, in addition to Y125, S129, Y133 and Y135 (*Oueslati et al., 2012*; *Paleologou et al., 2010*). S87 is the only phosphorylation site located within the NAC region. The previous solid-state NMR structure of α-Syn placed the side chain of this residue towards the inside of the protofilament core, leading to the assumption that phosphorylation of S87 might be the only modification occurring at a region not accessible in the fibrillar state. However, in our cryo-EM structure, S87 faces the outside of the fibril and hence remains accessible for disease-associated modification in α-Syn fibrils.

We also observed the arrangement of G47 and A78 described by *Tuttle et al. (2016)*, which was proposed to favor the interaction between residues E46 and K80 and allow them to form a stable salt bridge between two consecutive α-Syn monomers (*Figure 1—figure supplement 3G*). The conservation of the geometry adopted by these residues confirms their role in facilitating backbone-backbone interactions. In addition, our structure also confirms that residues A69 and G93 (and likely G68) help to stabilize the distal loop in a protofilament (*Figure 1—figure supplement 3F*).

A microED structure obtained from crystals produced from a 10-residue peptide simulating the core of α-Syn fibrils (PreNAC, from 47 to 56; *Figure 4B*) and including a threonine instead of an alanine at position 53 (i.e. A53T), also proposed that residue 53 forms the hydrophobic core within a protofilament (*Rodriguez et al., 2015*). In addition, the microED model suggested that the interaction between adjacent protofilaments would occur through residues 68 to 78 (referred to as NACore) (*Rodriguez et al., 2015*). However, their short peptides did not include most residues responsible for the α-Syn monomer topology that we observed. Instead, our cryo-EM structure reveals that the PreNAC is responsible for the interaction between protofilaments, and places the NACore at the very center (i.e.the core) of a single protofilament.

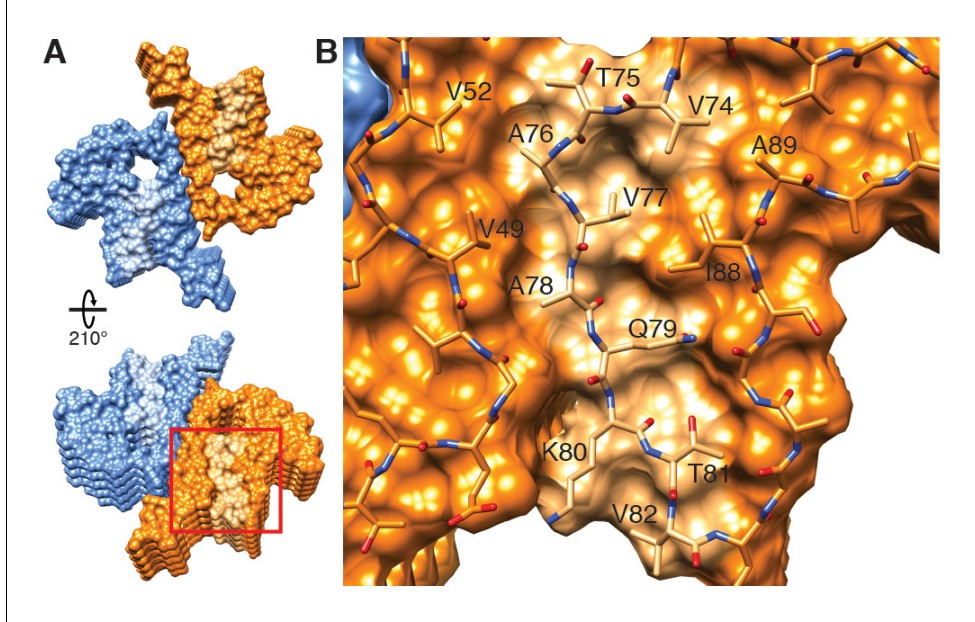

**Figure 5.** Hydrophobic cleft at the growing end of α-Syn(1-121) fibrils. (A) Views of opposite ends of α-Syn fibrils with the two protofilaments colored orange and blue. Regions corresponding to the location of the hydrophobic cleft are shown in a lighter shade. (B) Residues forming the hydrophobic cleft, including V49, V52, I88, A89 provide an entry point for residues V74-V82 of an incoming α-Syn molecule (atoms shown). Area shown in panel (B) is marked in panel (A) with a square.
DOI: https://doi.org/10.7554/eLife.36402.012

## Possible mechanism of fibril elongation

Our 3D structure allows us to hypothesize a mechanism for fibril elongation (fibril growth). Because two different stacking modes are present (i.e. the half-stack at the intermolecular interface and the stacking of β-strand β6), the two ends of the fibrils are distinct, suggesting an end-dependent growth of the fibrils, as documented and also suggested for other amyloids (*Lührs et al., 2005*). One end of the fibril includes a hydrophobic cleft formed between β-strands β2/ β3 on one side and β7 on the other side (residues V49, V52, A88, I89), providing a hydrophobic entry point for the next incoming molecule, with the matching segment consisting of 5 hydrophobic residues (V74-V82, *Figure 5*). This suggests that the initial binding event of fibril elongation might be a hydrophobic interaction involving residues V74-V82. This peptide segment is the central part of the NAC region and strong experimental evidence suggests that it is critical for fibril formation (*Giasson et al., 2001*). In addition, it has been shown that β-synuclein, which lacks residues V74 to V82, is incapable of forming fibrils (*Giasson et al., 2001*).

It is intriguing to speculate that a small molecule binding into this hydrophobic cleft could be a potent fibril elongation inhibitor or tracer, with the potential to be applied in PD and other synucleinopathies. Finally, the inter-molecular stacking may also play a role in fibril elongation, since the zipper interaction is of hydrophobic nature. Furthermore, it is likely that fibril growth alternates between the two protofilament structures at the level of monomer addition. Failure thereof may result in the growth of a single protofilament with little stability, yielding a dynamic on- and off-binding of monomers and larger oligomers, which has been observed for other amyloid fibril systems (*Carulla et al., 2005*).

In conclusion, we present the structure of recombinant α-Syn(1-121) fibrils determined at a resolution of 3.4 Å by cryo-EM. Our structure encompasses nearly the complete protein (residues 38 to 95), and includes the NAC region (residues 61 to 95) of α-Syn. We determined that various residues associated with familial forms of PD and other synucleinopathies are located in the interacting region between two protofilaments, suggesting their involvement in fibril formation and stabilization. The cryo-EM structure presented here reveals how two protofilaments interact to form a fibril, and how the NAC region contributes to protofilament formation and stability. Our structure also presents novel insights into how several PD-relevant mutations of α-Syn would compromise the structure of this fibril, suggesting that in the case of certain familial forms of PD, a different structure of α-Syn than this fibril strain might be involved. Our findings on protofilament interaction and our hypothesis on the mechanism of fibril elongation invite for the design of molecules for diagnostics or treatment of synucleinopathies.

# Materials and methods

## Recombinant proteins

Recombinant full-length α-Syn was expressed from the pRT21 expression vector in BL21(DE3) competent *Escherichia coli* (*E. coli*). For N-terminal acetylation of α-Syn, cells were pre-transfected by pNatB vector coding for the N-terminal acetylase complex (plasmid kindly provided by Daniel Mulvihill, School of Biosciences, University of Kent, Canterbury, UK) (*Johnson et al., 2010*). C-terminally truncated forms of α-Syn(1-119), α-Syn(1-121), and α-Syn(1-122) were expressed in BL21-DE3-pLysS competent *E. coli*. Purification of α-Syn strains was performed by periplasmic lysis, ion exchange chromatography, ammonium sulfate precipitation, and gel filtration chromatography as previously described (*Huang et al., 2005*; *Luk et al., 2009*). Polo like kinase 2 (PLK2) was expressed in BL21-DE3-pLysS competent *E. coli*, isolated via its His-tag and immediately used to phosphorylate purified α-Syn. This was followed by standard ion exchange and gel filtration chromatography to separate phosphorylated from non-phosphorylated α-Syn. Endotoxins were removed from all α-Syn strains by Detoxi-Gel Endotoxin Removing Gel (Thermo Scientific) usually in one run or until endotoxin levels were below detection level. The sequence of the expressed α-Syn strains was verified by tryptic digestion followed by MALDI mass spectrometry (MS) or HPLC/ESI tandem MS for total mass was performed. Purity and monodispersity was determined by Coomassie blue or Silver staining of the SDS PAGE gel and analytical ultracentrifugation and the concentration was determined by the bicinchoninic acid (BCA) assay (Thermo Scientific) with bovine serum albumin as a standard. Dialyzed and lyophilized α-Syn(1-121) was prepared by dialyzing the purified protein in a 2 kD Slide-A-Lyzer unit

**Table 1.** Cryo-EM structure determination and model statistics.

| | |
|---|---|
| **Data collection** | |
| Magnification | 165000 x |
| Pixel size (Å) | 0.831 |
| Defocus Range (μm) | −0.8 to −2.5 |
| Voltage | 300 kV |
| Exposure time (s per frame) | 0.2 |
| Number of frames | 50 |
| Total dose (e/Å$^2$) | 69 to 128 |
| **Reconstruction** | |
| Box size (pixels) | 280 |
| Inter-box distance (pixels) | 28 |
| Micrographs | 118 |
| Manually picked fibrils | 792 |
| Initial extracted segments | 18860 |
| Segments after 2D classification | 18371 |
| Segments after 3D classification | 13390 |
| Resolution after 3D refinement (Å) | 3.8 |
| Final resolution (Å) | 3.42 |
| Estimated map sharpening $B$-factor (Å$^2$) | −82.6 |
| Helical rise (Å) | 2.45 |
| Helical twist (°) | 179.5 |
| **Atomic model** | |
| Initial model used (PDB code) | 2N0A |
| Model resolution (Å)<br>FSC threshold | 2.94/4.08<br>FSC = 0.143/FSC = 0.5 |
| Model resolution range (Å) | 116.34–2.94 |
| Map sharpening $B$-factor (Å$^2$) | −82.6 |
| Model composition<br>Non-hydrogen atoms<br>Protein residues<br>Ligands | <br>3960<br>580<br>0 |
| $B$-factors (Å$^2$) (non-hydrogen atoms)<br>Protein<br>Ligand | <br>29.85<br>N.A. |
| R.m.s. deviations<br>Bond lengths (Å)<br>Bond angles (°) | <br>0.008<br>1.088 |
| Validation<br>MolProbity score<br>Clashscore<br>Poor rotamers (%) | <br>1.49<br>1.24<br>0.00 |
| Ramachandran plot<br>Favored (%)<br>Allowed (%)<br>Disallowed (%) | <br>85.71<br>14.29<br>0.00 |

DOI: https://doi.org/10.7554/eLife.36402.013

(Thermo Scientific, for max. 3 ml) against HPLC-water (VWR). 500 μg protein aliquots were pipetted into 1.5 ml tubes, frozen on dry ice, and lyophilized for 2 hr using an Eppendorf concentrator (Eppendorf). Lyophilized samples were stored at −80°C until use.

## Fibrillization

Fibrils were prepared by dissolving dialyzed and lyophilized, recombinant α-Syn protein at 5 mg/mL in incubation buffer (DPBS, Gibco; 2.66 mM KCL, 1.47 mM $KH_2PO_4$, 137.93 mM NaCl, 8.06 mM $Na_2HPO_4$-$7H_2O$ pH 7.0–7.3). Reactions of 200 µL per tube were incubated at 37°C with constant agitation (1,000 rpm) in an orbital mixer (Eppendorf). Reactions were stopped after 5 days, sonicated (5 min in a Branson 2510 water bath), aliquoted, and stored at −80°C until use. The presence of amyloid fibrils was confirmed by thioflavin T fluorimetry and high molecular weight assemblies were visualized by gel electrophoresis.

## Electron microscopy

Cryo-EM grids were prepared using a Vitrobot Mark IV (ThermoFisher Scientific) with 95% humidity at 4°C. Amyloid fibrils (3 µL aliquots) were applied onto glow-discharged, 300 mesh, copper Quantifoil grids. After blotting, grids were plunge frozen in liquid ethane cooled by liquid nitrogen. Samples were imaged on a Titan Krios (ThermoFisher Scientific) transmission electron microscope, operated at 300 kV and equipped with a Gatan Quantum-LS imaging energy filter (GIF, 20 eV energy loss window; Gatan Inc.). Images were acquired on a K2 Summit electron counting direct detection camera (Gatan Inc.) in dose fractionation mode (50 frames) using the Serial EM software (*Mastronarde, 2005*) at a magnification of 165,000× (physical pixel size 0.831 Å) and a total dose of ~69 electrons per square angstrom ($e^-$/$Å^2$) for each micrograph. Micrographs were drift-corrected and dose-weighted using MotionCor2 (*Zheng et al., 2017*) through the Focus interface (*Biyani et al., 2017*). Additional data collection parameters are detailed in *Table 1*.

## Image processing

Helical reconstruction was carried out with the RELION 2.1 software (*Scheres, 2012*), using methods described in *He and Scheres (2017)*. Filaments were manually selected using the helix picker in RELION 2.1. Filament segments were extracted using a box size of 280 pixels (233 Å) and an interbox distance of 28 pixels. A total of 18,860 segments were extracted from 792 fibrils manually picked from 118 micrographs (*Table 1*). 2D classification was carried out with a regularization value of T = 10, and 2D class averages with a clear separation of β-strands were selected for further data processing. Power spectra of 2D class averages show the layer line at 1/ (4.9 Å) with peak intensities on both sides of the meridian (Bessel order n = 1). This is the result of an approximate $2_1$ screw symmetry between α-Syn subunits on the two protofilaments (*Figure 1—figure supplement 2*). Segments assigned to the best 2D classes were used for 3D classification using a regularization value of T = 8 and with optimization of the helical twist and rise. For both 3D classification and refinement, a *helical_z_percentage* parameter of 10% was used, which defines the size of the central part of the intermediate asymmetrical reconstruction that is used to apply real-space helical symmetry (*He and Scheres, 2017*). An initial reconstruction was calculated using a cylinder generated via the helix toolbox in RELION 2.1 as initial model. This reconstruction was low-pass filtered to 60 Å and employed as the initial model for a 3D classification with a single class (K = 1) and T = 20, an approach that allowed the successful reconstruction of amyloid filaments (*Fitzpatrick et al., 2017*). The handedness of the reconstruction was determined by comparison with atomic force microscopy images, which showed left-coiled surface patterns for the fibrils.

Refinement was carried out by the auto-refine procedure with optimization of helical twist and rise. This resulted in a structure with overall resolution of 3.8 Å. Post-processing with a soft-edge mask and an estimated map sharpening *B*-factor of −82.6 Å gave a map with a resolution of 3.4 Å (by the FSC 0.143 criterion). An estimation of local resolution was obtained using RELION 2.1 and a local-resolution-filtered map was calculated for model building and refinement.

## Model building and refinement

A model of the α-Syn(1-121) fibril was built into the Relion local resolution-filtered map using COOT (*Emsley and Cowtan, 2004*), with the PDB ID 2N0A as an initial model for the early interpretation of the map. The structure helped to determine the directionality of the protein chain and facilitated the assignment of densities in the map to specific residues. However, due to the large differences between the NMR structure and our EM map, major rebuilding was necessary. The high quality of the EM map allowed us to unambiguously build residues 38–95. A comparison was also carried out

between our structure and X-ray structures of α-Syn fragments 69–77 (PDB ID 4RIK), 68–78 (PDB ID 4RIL) and 47–56 (PDB ID 4ZNN; with the mutation A53T).

The structure (10 monomers, 5 on each protofilament) was refined against the RELION local resolution-filtered map with PHENIX real space refine (*Afonine et al., 2013*). Rotamer, Ramachandran restraints, and 'NCS' constraints were imposed, and two *B*-factors per residue were used during refinement. For validation, we randomized the coordinates (with a mean shift of 0.3 Å) and refined (using the same settings) against one of the refinement half-maps (half-map 1). We then calculated the FSC between that model (after refinement against half-map 1) and half-map 1, as well as the FSC between the same model and half-map 2 (against which it was not refined). The lack of large discrepancies between both FSC curves indicates no overfitting took place.

## Acknowledgements

We thank Liz Spycher, Jana Ebner, Alexandra Kronenberger, Daniel Schlatter, Daniela Huegin, Ralph Thoma, Christian Miscenic, Martin Siegrist, Sylwia Huber, Arne Rufer, Eric Kusznir, Peter Jakob, Tom Dunkley, Joerg Hoernschmeyer, and Johannes Erny at Roche for their technical support to clone, express, purify and characterize the different recombinant forms of α-Syn, Kenneth N Goldie, Lubomir Kovacik and Ariane Fecteau-Lefebvre for support in cryo-EM, Shirley A Müller for support in manuscript preparation, and Sjors Scheres for help in image processing. Calculations were performed using the high-performance computing (HPC) infrastructure administered by the scientific computing center at University of Basel (sciCORE; http://scicore.unibas.ch). Plasmids for C-terminally truncated alpha-synuclein courtesy of Prothena, South San Francisco, CA, USA. This work was in part supported by the Synapsis Foundation Switzerland, and the Swiss National Science Foundation (grants CRSII3_154461 and CRSII5_177195). The authors declare no competing financial interests.

## Additional information

### Competing interests

Daniel Mona, Matthias E Lauer, Markus Britschgi: Employed by Hoffmann-La Roche. The other authors declare that no competing interests exist.

### Funding

| Funder | Grant reference number | Author |
|---|---|---|
| Schweizerischer Nationalfonds zur Förderung der Wissenschaftlichen Forschung | CRSII3_154461 | Ricardo Guerrero-Ferreira Nicholas MI Taylor |
| Stiftung Synapsis - Alzheimer Forschung Schweiz AFS | | Henning Stahlberg |
| Schweizerischer Nationalfonds zur Förderung der Wissenschaftlichen Forschung | CRSII5_177195 | Ricardo Guerrero-Ferreira Nicholas MI Taylor |

The funders had no role in study design, data collection and interpretation, or the decision to submit the work for publication.

### Author contributions

Ricardo Guerrero-Ferreira, Conceptualization, Data curation, Formal analysis, Validation, Investigation, Visualization, Methodology, Writing—original draft, Writing—review and editing; Nicholas MI Taylor, Software, Validation, Visualization, Writing—review and editing; Daniel Mona, Philippe Ringler, Matthias E Lauer, Investigation, Writing—review and editing; Roland Riek, Formal analysis, Investigation, Writing—original draft, Writing—review and editing; Markus Britschgi, Resources, Formal analysis, Investigation, Writing—review and editing; Henning Stahlberg, Conceptualization, Resources, Supervision, Funding acquisition, Investigation, Methodology, Writing—original draft, Project administration, Writing—review and editing

## Author ORCIDs

Ricardo Guerrero-Ferreira (iD) http://orcid.org/0000-0002-3664-8277
Nicholas MI Taylor (iD) https://orcid.org/0000-0003-0761-4921
Philippe Ringler (iD) https://orcid.org/0000-0003-4346-5089
Matthias E Lauer (iD) https://orcid.org/0000-0003-3252-8718
Markus Britschgi (iD) http://orcid.org/0000-0001-6151-4257
Henning Stahlberg (iD) http://orcid.org/0000-0002-1185-4592

## Decision letter and Author response

Decision letter https://doi.org/10.7554/eLife.36402.021
Author response https://doi.org/10.7554/eLife.36402.022

## Additional files

### Data availability

The cryo-EM image data are available in the Electron Microscopy Public Image Archive, entry number EMPIAR-10195. The 3D map is available in the EMDB, entry number EMD-0148. The atomic coordinates are available at the PDB, entry number PDB 6H6B.

The following datasets were generated:

| Author(s) | Year | Dataset title | Dataset URL | Database, license, and accessibility information |
|---|---|---|---|---|
| Guerrero-Ferreira R, Taylor N M I, Mona D, Riek R, Britschgi M, Stahlberg H | 2018 | Structure of alpha-synuclein fibrils | http://www.rcsb.org/structure/6h6b | Publicly available at the RCSB Protein Data Bank (accession no. 6H6B) |
| Guerrero-Ferreira R, Taylor N M I, Mona D, Riek R, Britschgi M, Stahlberg H | 2018 | Structure of alpha-synuclein fibrils | https://www.ebi.ac.uk/pdbe/entry/emdb/EMD-0148 | Publicly available at the European Nucleotide Archive (accession no: EMD-0148) |
| Guerrero-Ferreira R, Taylor N M I, Mona D, Riek R, Britschgi M, Stahlberg H | 2018 | Structure of alpha-synuclein fibrils | https://www.ebi.ac.uk/pdbe/emdb/empiar/entry/10195/ | Publicly available at the European Nucleotide Archive (accession no: EMPIAR-10195) |

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
