## [Decision Letter]

Thank you for submitting your article "Cryo-EM structure of alphα-Synuclein fibrils" for consideration by *eLife*. Your article has been reviewed by three peer reviewers, including Sjors HW Scheres as the Reviewing Editor and Reviewer #1, and the evaluation has been overseen by John Kuriyan as the Senior Editor. The following individuals involved in review of your submission have agreed to reveal their identity: Marcus Fändrich (Reviewer #2); Dennis J Selkoe (Reviewer #3).

The reviewers have discussed the reviews with one another and the Reviewing Editor has drafted this decision to help you prepare a revised submission.

Summary:

In this study, Guerrero-Ferreira and collaborators use helical reconstruction cryo-electron microscopy to report a near-atomic resolution structure of an α-synuclein (α-Syn) fibril. Unlike previous high-resolution structures of α-synuclein fibrils obtained with solid-state NMR and micro-ED, cryo-EM allowed the determination of an arrangement where two protofilaments form a polar fibril composed of β-strands. Interestingly, a region associated with mutations that predispose to familial forms of Parkinson's disease is in the interface between the two protofilaments, where the mutants could influence the formation and stability of fibrils.

Overall, this study is experimentally sound and provides further evidence of the power of cryo-EM to elucidate the structures of amyloid-like fibrils made from recombinant proteins in vitro. However, all three reviewers agreed that the current manuscript would need a very careful and major revision before being suitable for publication in *eLife*. Besides adequately addressing the important conceptual points outlined by the three reviewers below, the manuscript should also be subjected to an extensive rewrite as the language used is often unclear and imprecise.

There is some overlap between point 2 by reviewer #3 and the paragraph with the request for additional evidence of seeding by reviewer #1. After deliberation among the reviewers it was agreed that it would probably be hard to find convincing arguments for the pathophysiological relevance of the solved fibrils at this point. Therefore, the revision should at least address the comments made by reviewer #3, i.e. to better explain the choice made for the 1-121 mutant.

*Reviewer #1:*

This manuscript describes a 3.4Å cryo-EM structure of in vitro aggregated α-synuclein fibrils (from a construct comprising residues 1-121). The authors built a near-atomic model of the α-synuclein fibril, which contains two protofilaments with a helical pseudo-two-fold screw symmetry. Each protofilament comprises eight parallel β-strands, forming a hydrophobic core with disordered random-coil-like tails on both ends. The authors describe the position of several familial Parkinson's disease (PD) mutations in the context of the 3D fibril structure, interpreting that the mutations would compromise the formation of the fibril structure presented here. The structure shares a similar topology with an NMR structure of in-vitro aggregated full-length α-synuclein fibrils that form single protofilaments. The manuscript also proposes a possible mechanism of fibril growth based on the observation of a hydrophobic cleft on one side of the fibril, which would then also be a potential target for fibril growth inhibitors or tracer compounds.

Although the structure presented is interesting and could in principle be worthy of publication in *eLife*, the current manuscript is written very poorly. Moreover, there is no additional data to back up some of the claims made based on the structure. Therefore, this paper will need a very major revision before it will be suitable for publication.

The Introduction is very short, and does not do credit to the large body of literature existing on α-synuclein.

Being the result of an in-vitro aggregation experiment, the relevance of the structures in disease is questionable. This is further exacerbated by using a truncation construct instead of the wildtype protein. It is not entirely clear from Figure 1—figure supplement 1 why the authors did not actually perform structure determination of the wildtype protein. Because of this change, the experiments performed on wildtype protein before by others may not be relevant for this structure. Therefore, the authors should provide evidence that the fibrils they used can still seed aggregation in cell cultures or mouse models.

Subsection “The familial PD mutations in the context of the 3D fibril structure”, end of second paragraph: I cannot see how the observation that familial PD mutations lie at the interface between the protofilaments would suggest that this fibril has some sort of 'native function'. Then, that exact statement is refuted in the next line, where evolution towards a fibril with a hydrophobic patches is deemed unlikely. This second statement is also strange. Why would there be an evolutionary pressure to form fibrils in the first place, regardless of their hydrophobicity? And more importantly, what do the authors themselves think is the relevance of this structure?

The authors discuss differences with the solid-state NMR structure by Tuttle et al. It could be that the filaments are indeed different, but it is perhaps equally likely that the ssNMR model is wrong, as not enough restraints may have been acquired to define a unique structure. Therefore, it would be interesting to know whether the cryo-EM structure actually explains the NMR data, or whether some distances measured by NMR are incompatible with the cryoEM structure, and the two are indeed different. This possibility should at least be discussed.

Subsection “Possible mechanism of fibril growth”, first sentence: Now the fibrils are called 'the transmissible species' solely based on some similarities with the NMR structure (which is in fact rather different and comprises only a single protofilament). This is at least confusing, as previously it was suggested this fibril structure is different from the one relevant to PD.

The authors hypothesize a mechanism for fibril growth based on the observation of a cleft on one side of the fibril. Although interesting in principle, they refer to their 'findings on the mechanism of fibril elongation'. However, they present no additional data that actually provides proof of a single 'growing end' of the fibril, nor its directionality.

*Reviewer #2:*

The study is well done. However, it contains a rather serious error in the β-sheet nomenclature, and it requires some further clarifications regarding the analysis before publication can be recommended.

1) There is a misclassification of the β-sheet structure, which does *not* possess a Greek key topology. Richardson defined a Greek key as a (+3) -1 -1 (+3) topology of 5 hydrogen bonded strands from the SAME β-sheet, with the exception that the first or the last strand can be missing. In the present case, however, the strands belong to β-sheets that are clearly different, and the strands do not interact with one another through backbone hydrogen bonds but rather via their side chains. Unfortunately, this misclassification was introduced previously to the amyloid field by another paper, but it is now important that we now avoid an error propagation. The term Greek key is misleading in the present context and needs to be deleted.

2) Which features in the original cryo-EM images show that the fibril possesses a screw axis? Please clarify through a new figure in the supplementary information.

3) The manuscript currently does not describe well how the obtained 3D density corresponds to the original experimental data? This should also merit a figure.

4) Please present evidence that the model is an adequate description of the 3D map or original cryo-EM data, for example, by comparing class averages with 2D projections of the density or by comparisons of the power spectra obtained with class averages, 3D maps and models.

*Reviewer #3:*

In this study, Guerrero-Ferreira and collaborators use helical reconstruction cryo-electron microscopy to report a near-atomic resolution structure of an α-synuclein (α-Syn) fibril. Unlike previous high-resolution structures of α-synuclein fibrils obtained with solid-state NMR and micro-ED, cryo-EM allowed the determination of an arrangement where two protofilaments form a polar fibril composed of β-strands. Interestingly, a region associated with mutations that predispose to familial forms of Parkinson's disease is in the interface between the two protofilaments, where the mutants could influence the formation and stability of fibrils.

Overall, this study is experimentally sound, well-illustrated and provides further evidence of the power of cryo-EM to elucidate the structures of amyloid-like fibrils made from recombinant proteins in vitro.

While the structural data appear to be appropriately analyzed and illustrated, I will allow structural biologists to comment on the details of the analysis of the cryoEM images of the 2-protofilament, 10 nm fibril solved here. Regardless of the interpretation of the molecular details of this 3D fibril structure, there are some major conceptual issues (numbered below) about their report that the authors should address in order to support the relevance of this aa1-121 fibril generated in vitro to the normal in vivo biology of α-Syn and the role of α-Syn fibrils in human synucleinopathies, particularly PD.

1) Already in the Abstract and Introduction, the authors should explain to readers whether they view the fibrils they elegantly analyze by cryoEM as modeling normal or pathological assembles. This is not really made clear until a rather speculative discussion where they write (somewhat ambiguously) "this suggests that this fibril structure might have a native function, and that a different fibril structure could be involved in the toxicity of the disease". Then, in the next sentence, they state an apparently contradictory opinion: "it seems likely that the fibril structure presented here doesn't have a native or mechanistic function by itself", and shortly thereafter, they state that "similar, rather unique [meaning?] structural characteristics have been previously observed for tau". Are those tau structures believed by those respective authors and also by the current authors to be normal or pathological? It is widely assumed in the neurodegenerative field that α-Syn and tau are normally soluble and functional proteins in neurons, and that amyloid-like fibrils or protofilaments do not occur normally and are pathological. But in the Results and Discussion, and elsewhere in the manuscript, the authors seem unclear about the precise disease-relevance of this wild-type α-Syn fibril. An example of this ambiguity is the sentence: "The here presented structure therefore hints at the possibility that a different structure or mechanism might be the causative agent for PD, at least for familial PD". This sentence sounds vague, and the term "causative agent for PD" seems overreaching. Such statements may be confusing to α-Syn biologists trying to interpret the meaning and impact of this structural work.

2) At the beginning of Results, the authors show Ems of "several preparations of recombinant human α-Syn fibril", including those formed by full-length (FL) unmodified α-Syn and N-terminally acetylated α-Syn. They then say that "the fibrils formed by α-Syn(1-121) were straight, between 20 and 500 nm long and the only ones consistently 10 nm in diameter". They conclude "for this reason, α-Syn(1-121) was used to proceed with structural analysis by cryo-EM" For what reason? Why do these in vitro characteristics justify choosing the α-Syn(1-121) variant to focus their work on? This seems arbitrary or at least not well justified. The authors should explain this and further state why an N-acetylated FL α-Syn (filaments of which are nicely shown by EM in Figure 1C) would not have been more biologically relevant, since the vast majority of α-Syn in human neurons is in this state, both in healthy people and in PD subjects. An example of this concern arises in the first paragraph of the subsection “Comparison with earlier structural data”, where the authors contrast their new structure of α-Syn(1-121) to that of Tuttle et al. of a ssNMR structure of a pathogenic fibril of FL α-Syn; this contrast raises the question of why the authors' α-Syn(121) fibril is a more biologically relevant structure.

3) The authors state already in the Abstract and in several places throughout the paper that human PD has as a major feature the cell-to-cell propagation of α-Syn. While this is a popular and much studied hypothesis, there is no direct evidence to date that physical passage of α-Syn between neurons occurs – and is required for – in the human disease, i.e., occurs in patients with PD and DLB. This is especially true as regards fully formed amyloid-like fibrils of α-Syn, despite the authors' claim that "the fibril species presented here is likely to be the transmissible species." Even those championing the pathogenic spread hypothesis do not claim that fully formed, 10 nm fibrils such as are examined here are the likely form of α-Syn that spreads from neuron to neuron. More readily diffusible soluble oligomers of α-Syn are more likely to be candidates for inter-neuronal passage. The authors would be wise to reserve their speculations about this to the end of their paper, assuming they believe that neuron-to-neuron transport of the 10 nm fibrils they characterize will turn out be biologically true in PD patients, rather than assuming this outcome already in the second sentence of the Abstract and in the Introduction (third paragraph). The authors should also clarify for readers that their statement "our 3D structure reveals some detailed insight into the mechanism of fibril replication" is distinct from the mechanisms of any fibril propagation or "transmissibility", which they mention just 3 lines earlier. These terms ("replication" and "transmissibility") should not be conflated in the reader's mind.

4) A related question of biological interpretation and PD-relevance is whether the authors assume the 10 nm amyloid-type fibrils they define would occur intra- or extracellularly. If intraneuronal, do they then believe their fibrils are similar to the α-Syn filaments that accumulate in Lewy bodies and neurites? In this context, the authors emphasize that their fibrils are of wild-type α-Syn and that PD-causing mutations (e.g., H50Q, G51D, A53E, A53T) would each alter "the inter-molecular contact between two β-sheets from adjacent protofilaments at the core of the fibrils" they study, including the disruption of the stearic zipper interaction they postulate on the basis of their 3D structure. So, while an additional structure determination of an α-Syn mutant is probably beyond the scope of this study, it would be very interesting to at least have a rough a priori reconstruction of any of the fPD mutants occurring at the interface between the protofilaments – especially in light of their strong claims about the importance of this interface. Would the authors expect to observe a radically different structure?

5) The advance of Fitzpatrick and collaborators' determination of the structure of tau filaments derives from the fact that the material came from human patients. Would the authors expect fibrils within Lewy bodies in patients suffering from synucleinopathies to maintain the same overall structure they report here?

6) Compelling ssNMR data suggest the existence of only minor structural differences between wt α-synuclein and the A53T mutant. How would this affect the authors' assumption that the mutations are clustered at the core of the interacting protofilaments and therefore have major effects on the structure of the wt fibrils prepared here?

7) The α-synuclein fibril diameter reported by some labs within Lewy bodies is ~5 nm, akin to a single protofilament of the current structure. How do the authors justify the importance of a two-protofilament fibril made here from bacterially expressed, non-acetylated recombinant α-Syn?

8) Any comment on why the authors find that α-Syn(1-119) is unable to form fibrils (Figure 1D), despite the fact that the C-terminal residues remain dynamic even in the fibrillar state?

---

## [Author Response]

Reviewer #1:[…] Although the structure presented is interesting and could in principle be worthy of publication in eLife, the current manuscript is written very poorly. Moreover, there is no additional data to back up some of the claims made based on the structure. Therefore, this paper will need a very major revision before it will be suitable for publication.The Introduction is very short, and does not do credit to the large body of literature existing on α-synuclein.

The introduction has been extended to include a larger review of the existing α-synuclein literature.

Being the result of an in-vitro aggregation experiment, the relevance of the structures in disease is questionable. This is further exacerbated by using a truncation construct instead of the wildtype protein. It is not entirely clear from Figure 1—figure supplement 1 why the authors did not actually perform structure determination of the wildtype protein. Because of this change, the experiments performed on wildtype protein before by others may not be relevant for this structure. Therefore, the authors should provide evidence that the fibrils they used can still seed aggregation in cell cultures or mouse models.

In order to highlight the importance of this fibril form (α-Syn 1-121) and explain why it was selected for cryo-EM structure determination, we have added references to published work on the α-Syn 1-121 fibrils. Previous in vivo experiments with a neuronal Parkinson’s Disease cell model as well as a mouse model of Multiple System Atrophy showed the aggregation propensity of α-Syn 1-121, as well as its association with neuronal cell toxicity. A recombinant α-Syn 1-121 has previously been investigated as well and exhibited the same aggregation profile. This is now detailed in the manuscript.

Relevant references are listed in the manuscript. These include: Bassil et al., 2016; Wang et al., 2016.

Subsection “The familial PD mutations in the context of the 3D fibril structure”, end of second paragraph: I cannot see how the observation that familial PD mutations lie at the interface between the protofilaments would suggest that this fibril has some sort of 'native function'. Then, that exact statement is refuted in the next line, where evolution towards a fibril with a hydrophobic patches is deemed unlikely. This second statement is also strange. Why would there be an evolutionary pressure to form fibrils in the first place, regardless of their hydrophobicity? And more importantly, what do the authors themselves think is the relevance of this structure?

A description of the relevance of this structure has been added to the Results and Discussion section of the manuscript. In addition, the above-mentioned claims have been removed. The Discussion was extended and corrected.

The authors discuss differences with the solid-state NMR structure by Tuttle et al. It could be that the filaments are indeed different, but it is perhaps equally likely that the ssNMR model is wrong, as not enough restraints may have been acquired to define a unique structure. Therefore, it would be interesting to know whether the cryo-EM structure actually explains the NMR data, or whether some distances measured by NMR are incompatible with the cryoEM structure, and the two are indeed different. This possibility should at least be discussed.

We have now re-written the discussion of the NMR structure. We highlight the similarities (general fold) and the differences (single protofibril in NMR, double fibril in our map) The NMR structure had a significantly higher discrepancy between lowest-energy structures for residues 51-67 (Tuttle et al., Figure 3D), and exactly for these residues, our map has the side chains pointing into the opposite direction than in the NMR structure. We indicate this in the revised manuscript.

Subsection “Possible mechanism of fibril growth”, first sentence: Now the fibrils are called 'the transmissible species' solely based on some similarities with the NMR structure (which is in fact rather different and comprises only a single protofilament). This is at least confusing, as previously it was suggested this fibril structure is different from the one relevant to PD.

We agree with the reviewer. As cell-to-cell propagation of fibrils was not the scope of our study, the parts of the text referring to neuron-to-neuro transfer of α-Syn fibrils have now been removed.

The authors hypothesize a mechanism for fibril growth based on the observation of a cleft on one side of the fibril. Although interesting in principle, they refer to their 'findings on the mechanism of fibril elongation'. However, they present no additional data that actually provides proof of a single 'growing end' of the fibril, nor its directionality.

The text referring to the mechanism of fibril elongation was modified as suggested and is now clearly stated as a hypothesis.

Reviewer #2:The study is well done. However, it contains a rather serious error in the β-sheet nomenclature, and it requires some further clarifications regarding the analysis before publication can be recommended.1) There is a misclassification of the β-sheet structure, which does not possess a Greek key topology. Richardson defined a Greek key as a (+3) -1 -1 (+3) topology of 5 hydrogen bonded strands from the SAME β-sheet, with the exception that the first or the last strand can be missing. In the present case, however, the strands belong to β-sheets that are clearly different, and the strands do not interact with one another through backbone hydrogen bonds but rather via their side chains. Unfortunately, this misclassification was introduced previously to the amyloid field by another paper, but it is now important that we now avoid an error propagation. The term Greek key is misleading in the present context and needs to be deleted.

We thank the reviewer for the clarification of the terminology. As suggested, the term Greek key has been removed from the text.

2) Which features in the original cryo-EM images show that the fibril possesses a screw axis? Please clarify through a new figure in the supplementary information.

Additional panels were added to Figure 1—figure supplement 2 with 2D class averages and power spectra showing no meridional intensities on the 1/(4.9 Å) layer line with results from an approximate 2_1_ screw axis. An explanation was also added to the Image Processing section of the Materials and methods.

3) The manuscript currently does not describe well how the obtained 3D density corresponds to the original experimental data? This should also merit a figure.

Additional panels were added to Figure 1—figure supplement 2 to compare class averages with 2D projections of the 3D map.

4) Please present evidence that the model is an adequate description of the 3D map or original cryo-EM data, for example, by comparing class averages with 2D projections of the density or by comparisons of the power spectra obtained with class averages, 3D maps and models.

Additional panels were added to Figure 1—figure supplement 2 to compare class averages with 2D projections of the 3D map and the model as suggested. Power spectra for all sets of images were also included. The figure legend was modified accordingly.

Reviewer #3:[…] 1) Already in the Abstract and Introduction, the authors should explain to readers whether they view the fibrils they elegantly analyze by cryoEM as modeling normal or pathological assembles. This is not really made clear until a rather speculative discussion where they write (somewhat ambiguously) "this suggests that this fibril structure might have a native function, and that a different fibril structure could be involved in the toxicity of the disease". Then, in the next sentence, they state an apparently contradictory opinion: "it seems likely that the fibril structure presented here doesn't have a native or mechanistic function by itself", and shortly thereafter, they state that "similar, rather unique [meaning?] structural characteristics have been previously observed for tau". Are those tau structures believed by those respective authors and also by the current authors to be normal or pathological? It is widely assumed in the neurodegenerative field that α-Syn and tau are normally soluble and functional proteins in neurons, and that amyloid-like fibrils or protofilaments do not occur normally and are pathological. But in the Results and Discussion, and elsewhere in the manuscript, the authors seem unclear about the precise disease-relevance of this wild-type α-Syn fibril. An example of this ambiguity is the sentence: "The here presented structure therefore hints at the possibility that a different structure or mechanism might be the causative agent for PD, at least for familial PD". This sentence sounds vague, and the term "causative agent for PD" seems overreaching. Such statements may be confusing to α-Syn biologists trying to interpret the meaning and impact of this structural work.

We are thankful for these observations, which allowed us to improve our Discussion.

The relevance of the solved structure has been explained in the Results and Discussion section of the manuscript. In addition, the Discussion has been corrected to better explain our points of view and clarify several sentences and the impact of our work.

2) At the beginning of Results, the authors show Ems of "several preparations of recombinant human α-Syn fibril", including those formed by full-length (FL) unmodified α-Syn and N-terminally acetylated α-Syn. They then say that "the fibrils formed by α-Syn(1-121) were straight, between 20 and 500 nm long and the only ones consistently 10 nm in diameter". They conclude "for this reason, α-Syn(1-121) was used to proceed with structural analysis by cryo-EM" For what reason? Why do these in vitro characteristics justify choosing the α-Syn(1-121) variant to focus their work on? This seems arbitrary or at least not well justified. The authors should explain this and further state why an N-acetylated FL α-Syn (filaments of which are nicely shown by EM in Figure 1C) would not have been more biologically relevant, since the vast majority of α-Syn in human neurons is in this state, both in healthy people and in PD subjects. An example of this concern arises in the first paragraph of the subsection “Comparison with earlier structural data”, where the authors contrast their new structure of α-Syn(1-121) to that of Tuttle et al. of a ssNMR structure of a pathogenic fibril of FL α-Syn; this contrast raises the question of why the authors' α-Syn(121) fibril is a more biologically relevant structure.

In order to highlight the importance of this fibril form (α-Syn 1-121) and extend on the reasons why it was picked for cryo-EM analysis, we have added text citing previous research concerning the α-Syn(1-121) truncation. in vivo experiments on a neuronal Parkinson’s Disease cell model as well as a mouse model of Multiple System Atrophy have demonstrated the aggregation propensity of this α-Syn species as well as its association with neuronal cell toxicity. We also explain that α-Syn(1-121) was the most suitable sample for a high-resolution structure analysis, due to the constant diameters and straight fibril structure.

3) The authors state already in the Abstract and in several places throughout the paper that human PD has as a major feature the cell-to-cell propagation of α-Syn. While this is a popular and much studied hypothesis, there is no direct evidence to date that physical passage of α-Syn between neurons occurs – and is required for – in the human disease, i.e., occurs in patients with PD and DLB. This is especially true as regards fully formed amyloid-like fibrils of α-Syn, despite the authors' claim that "the fibril species presented here is likely to be the transmissible species." Even those championing the pathogenic spread hypothesis do not claim that fully formed, 10 nm fibrils such as are examined here are the likely form of α-Syn that spreads from neuron to neuron. More readily diffusible soluble oligomers of α-Syn are more likely to be candidates for inter-neuronal passage. The authors would be wise to reserve their speculations about this to the end of their paper, assuming they believe that neuron-to-neuron transport of the 10 nm fibrils they characterize will turn out be biologically true in PD patients, rather than assuming this outcome already in the second sentence of the Abstract and in the Introduction (third paragraph).

We agree with the reviewer that mention of transmissibility of α-Syn fibrils may prove confusing. Determination of fibril cell-to-cell propagation was out of the scope of our study. Therefore, parts of the text referring to neuron-to-neuro transfer of α-Syn fibrils have been removed.

The authors should also clarify for readers that their statement "our 3D structure reveals some detailed insight into the mechanism of fibril replication" is distinct from the mechanisms of any fibril propagation or "transmissibility", which they mention just 3 lines earlier. These terms ("replication" and "transmissibility") should not be conflated in the reader's mind.

The term “transmissibility” has been removed from this section for clarity. We describe our proposed hypothesis on the mechanism of fibril elongation.

4) A related question of biological interpretation and PD-relevance is whether the authors assume the 10 nm amyloid-type fibrils they define would occur intra- or extracellularly. If intraneuronal, do they then believe their fibrils are similar to the α-Syn filaments that accumulate in Lewy bodies and neurites? In this context, the authors emphasize that their fibrils are of wild-type α-Syn and that PD-causing mutations (e.g., H50Q, G51D, A53E, A53T) would each alter "the inter-molecular contact between two β-sheets from adjacent protofilaments at the core of the fibrils" they study, including the disruption of the stearic zipper interaction they postulate on the basis of their 3D structure. So, while an additional structure determination of an α-Syn mutant is probably beyond the scope of this study, it would be very interesting to at least have a rough a priori reconstruction of any of the fPD mutants occurring at the interface between the protofilaments – especially in light of their strong claims about the importance of this interface. Would the authors expect to observe a radically different structure?

We appreciate the comment from this reviewer regarding the fibril structure of α-Syn with PD-causing mutations, and we do share the interest in solving such structures. However, structure determination of mutant α-Syn fibrils is likely a more difficult and long process, which we are afraid is beyond the scope of this study.

5) The advance of Fitzpatrick and collaborators' determination of the structure of tau filaments derives from the fact that the material came from human patients. Would the authors expect fibrils within Lewy bodies in patients suffering from synucleinopathies to maintain the same overall structure they report here?

Lewy bodies contain a high concentration of α-Syn. To our knowledge, purification of α-Syn fibrils from human brain so far requires rather harsh conditions (e.g., solubilization with sarcosine at elevated temperatures for several hours under stirring), and so far has not resulted in fibril material of sufficient quality for structural analysis. The fact that our here presented α-Syn fibril structure appears to be incompatible with certain familial PD α-Syn mutations, suggests that at least in some PD patients affected by those mutations, the here presented fibril structure should not be able to form.

6) Compelling ssNMR data suggest the existence of only minor structural differences between wt α-synuclein and the A53T mutant. How would this affect the authors' assumption that the mutations are clustered at the core of the interacting protofilaments and therefore have major effects on the structure of the wt fibrils prepared here?

α-Syn is known to form fibrils of different strain types (e.g., Peelaerts et al., 2015). Lemkau et al. (PLOS One, 2013) saw only local differences around residue 53 between wt and A53T fibrils. At this point, we cannot explain their findings, which were obtained by seeding and longer incubation times.

7) The α-synuclein fibril diameter reported by some labs within Lewy bodies is ~5 nm, akin to a single protofilament of the current structure. How do the authors justify the importance of a two-protofilament fibril made here from bacterially expressed, non-acetylated recombinant α-Syn?

There are reports of 10 nm fibers extracted from Lewy bodies (Crowther, Daniel, and Goedert, Neuroscience Letters (2000)). In their research, filaments were isolated from pooled substantia nigrae of 10 PD cases. Their research also suggests that the 10 nm filaments may contain 5 nm protofilaments as substructures.

A discussion of various instances where 5 and 10 nm fibrils have been found in brain of PD patients has been added as part of the Results and Discussion section for clarification.

8) Any comment on why the authors find that α-Syn(1-119) is unable to form fibrils (Figure 1D), despite the fact that the C-terminal residues remain dynamic even in the fibrillar state?

In our hands, the construct α-Syn(1-119) did not, or only form very short fibrils. We cannot explain this behavior. It may have something to do with shielding of fibril surface areas by the longer N- and C-termini.